# Diameter-Change-Induced Transition in Buckling Modes of Defective Zigzag Carbon Nanotubes

**DOI:** 10.3390/nano12152617

**Published:** 2022-07-29

**Authors:** Yoshitaka Umeno, Atsushi Kubo, Chutian Wang, Hiroyuki Shima

**Affiliations:** 1Institute of Industrial Science, The University of Tokyo, 4-6-1, Komaba, Tokyo 153-8505, Japan; kubo@ulab.iis.u-tokyo.ac.jp (A.K.); yakumojyuuni@gmail.com (C.W.); 2Department of Environmental Sciences, University of Yamanashi, 4-4-37, Takeda, Yamanashi 400-8510, Japan

**Keywords:** carbon nanotube, column buckling, shell buckling, Stone-Wales defect, axial compression, nanomechanics

## Abstract

In general, the insertion of Stone-Wales (SW) defects into single-walled carbon nanotubes (SWNTs) reduces the buckling resistance of SWNTs under axial compression. The magnitude of reduction is more noticeable in zigzag-type SWNTs than armchair- or chiral-type SWNTs; however, the relation between the magnitude of reduction and aspect ratio of the zigzag SWNTs remains unclear. This study conducted molecular dynamics (MD) simulation to unveil the buckling performance of zigzag SWNTs exhibiting SW defects with various tube diameter. The dependencies of energetically favorable buckling modes and the SW-defect induced reduction in the critical buckling point on the tube diameter were investigated in a systematic manner. In particular, an approximate expression for the critical buckling force as a function of the tube diameter was formulated based on the MD simulation data.

## 1. Introduction

A slender member often exhibits axial buckling under compression, which is a sudden archwise deflection of elastic materials with a large aspect ratio. It is a type of mechanical bifurcation that is triggered shortly after the axially compressive load exceeds a threshold value [1]. It is a universal phenomenon that occurs regardless of the length scale of the material, ranging from macroscopic to micro- and nano-sized materials, provided it is sufficiently slender. Macroscopic elastic materials have been extensively studied in terms of their axial buckling performance, as primarily described by the relation of the critical buckling stress with the slenderness ratio of the materials. In contrast, studies on buckling behavior of nano- and micro-sized elastic materials are scarce, partly because in addition to the slenderness ratio, it can be sensitive to the symmetry and irregularity of the atomic structure as well.

A carbon nanotube (CNT) is a typical example of nanomaterials endowed with a large aspect ratio. Owing to their superior mechanical flexibility and reversibility, CNTs exhibit axial buckling under compression [2,3], which has been experimentally observed in microscope experiments [4,5,6,7,8]. Meanwhile, sample manipulation on the nanometer scale remains a challenging task despite state-of-the-art microscopy techniques. This hindrance is particularly noticeable when dealing with single-walled carbon nanotubes (SWNTs), which are expected to show nontrivial correlation between mechanical deformation and their physical properties [9,10]. This may be a reason why tremendous efforts based on numerical simulations [10,11,12,13,14,15,16,17,18] and continuum approximation theory [11,17,19,20,21,22] have been exerted in exploring the buckling behavior of SWNTs under loading. Several theoretical studies have revealed that axial buckling of SWNTs is highly dependent on the lattice irregularity [23,24] and tube chirality [25,26,27]. This is in addition to the aspect ratio [11,26,28,29,30,31,32] similar to macroscopic tubular structures.

Among the several types of lattice irregularities, Stone-Wales (SW) defects are those peculiar to graphene layers with hexagonal symmetry [33]. An SW defect comprises two pentagon-heptagon pairs of carbon atoms, formed by local rearrangement of the original four adjacent hexagons [34]. It has been numerically proven that the presence of SW defects results in pronounced reductions in the buckling stress of SWNTs under compression [27,29,35,36,37] as well as their fracture stress under tension [38,39]. In particular, zigzag SWNTs exhibit a significant reduction in buckling stress caused by SW-defect insertion than armchair SWNTs, as confirmed in the comparative study of (17,0) zigzag SWNTs and (10,10) armchair SWNTs [27]. This chirality dependence is attributed to the difference in the relative angle of the central C-C bond to the tube axis. Considering that the critical buckling stress of tubular materials generally depends on its aspect ratio, the manner in the tube diameter variation in zigzag SWNTs exhibiting SW defects alter their buckling performance must be understood. However, studies to systematically investigate this issue are yet to be conducted.

This study investigated the axial buckling of zigzag SWNTs with various tube diameter using molecular dynamics (MD) simulations. SWNTs were found to exhibit three different buckling modes depending on the aspect ratio, regardless of the presence of a SW defect. In addition, the force-strain curve for each SWNTs having different tube diameters were computed. Consequently, the degree of reduction in the critical buckling force caused by SW-defect insertion was exhibited. Furthermore, an approximate function describing the dependence of the critical buckling force on the tube diameter both for defective and defect-free SWNTs was formulated.

## 2. Method

### 2.1. SWNT Model with SW Defect

The atomistic model of zigzag SWNTs was constructed under the following conditions. As per conventions, zigzag SWNTs with different tube radii are classified based on the index *n* that characterizes the chiral vector (n,0) of the tube. Using the index *n*, the tube diameter *d* of pristine zigzag SWNTs is described by $d=(\sqrt{3}a/\pi )n≃0.0783n$ nm with the assumption that the C-C bond length, *a*, at the undeformed state is a=0.142 nm. This study examined the buckling behavior of (n,0) zigzag SWNT models with 10≤n≤34, and all the models consist of 60 unit cells aligned along the axial direction with an initial length of 12.78 nm. The corresponding values of the aspect ratio (i.e., the ratio of the tube length to the tube diameter) was in the range of 4.81 to 16.3. Further, in actual calculations, the periodic boundary condition was applied in the axial direction of each SWNT model. In principle, the energetically optimized axial length of a SWNT may vary depending on *n*, due to the effect of curvature (rolling-up) of the graphene sheet on the bond lengths and angles. Yet in this study, the curvature effect was not taken into account since we have numerically confirmed that it is considerably small.

For each pristine SWNT model, a SW defect was created through the rotation of a C-C bond vertical to the tube axis by π/2 with no dangling bonds. Subsequently, the formation of the pentagon-heptagon pairs altered the geometry of the local area surrounding the defect. In fact, following energy minimization, the diameter of the defective area was found to increase slightly. When the SWNT was subjected to axial loading, the initial imperfection caused by the SW defect expedited the occurrence of the buckling.

However, SW defects are not always artificial imperfections and can be commonly generated during synthesis of CNTs [40,41]. Furthermore, tensile [42,43] and compressive [29] load application can produce SW defects even in pristine CNTs. Thus, considering the ubiquity of SW defects in synthesized CNTs, the effect of SW defects on the buckling performance of zigzag SWNTs with various tube diameters must be quantified.

### 2.2. Molecular Dynamics Simulation

The SWNT models were subjected to MD simulations wherein axial compression was applied. The atomic interaction between carbon atoms was represented by the reactive empirical bond order (REBO) potential [44], which has been widely used for hydrocarbon systems. The compressive strain applied to the model was increased stepwise. Further, to avoid the system being trapped in a metastable state (i.e., the state wherein no buckling deformation occurs, although the strain has exceeded the critical value for buckling), simulations were conducted performing short-time MD simulations and structural relaxations alternatively based on the global convergent algorithm [45]. In practice, we gradually increased the compressive strain stepwise, repeating the following three processes:The axial cell size is reduced by 0.01 nm, which corresponds to a compressive nominal strain of 0.078%, to increase the compressive strain. In this process, the fractional coordinates of the atoms are fixed so that the SWNT model undergoes uniform deformation;To prevent the system from falling into a state of unstable equilibrium, a short MD calculation of 10 fs at a temperature of 50 K is performed, where the MD time step is 1 fs;Structural relaxation (optimization of atom positions) is performed until all force vectors exerted on the atoms are smaller than $10^{-7}$ eV per angstrom. For the relaxation we adopted the GLOC algorithm [45].

All the simulations were performed using in-house software “MDSPASS2”.

## 3. Three Buckling Modes of Zigzag SWNTs

Figure 1 shows the axially buckling modes of zigzag SWNTs with and without SW defects. The models depicted in the upper panels possessed no defect, whereas those in the lower panels contained a single SW defect at the position marked by red arrows. The color of each constituent atom represents the magnitude of the local strain energy within a particular SWNT model. The energy at the red (or blue) atom was relatively higher (lower) than that at other positions; the energy scale was not fixed to be a constant across all panels and thus varies from panel to panel.

Three distinct modes were observed in Figure 1 depending on the index *n*. This implied that the buckling mode was strongly dependent on the tube diameter. For a small *n* value, the SWNTs underwent columnar buckling, wherein the tube deflected into a smooth *S*-like shape with no kink; hereafter, this buckling mode is referred to as the “*S*-mode”. Figure 1a,d show the *S*-mode of a (12,0) SWNT having no and one SW defect, respectively. For the defect-free SWNT, the edge-to-edge length shrank from the initial value to 12.43 nm immediately after buckling. Moreover, no clear critical point was found in the case of the defective model. In the S-mode of the defective SWNT, the location of the SW defect coincides with the position at which the generating line of the envelope cylindrical surface has a maximum curvature, as depicted in Figure 1d. Similar S-modes were found to occur under conditions of 10≤n≤13 for defect-free SWNTs, whereas they occurred under conditions of 10≤n≤14 for defective SWNTs.

When the value of *n* increased, the buckling mode switched from the *S* mode to another mode showing “*Z*-shaped” tube geometry. Figure 1b,e show the *Z*-mode of the (18,0) SWNT models without and with a SW defect, respectively. The edge-to-edge length immediately after buckling was estimated to be 12.16 nm (defect-free) and 12.24 nm (defective). Further, the *Z*-mode was endowed with two kinks, implying that this mode is categorized into shell buckling modes. Moreover, *Z*-modes were observed under conditions of 14≤n≤23 for defect-free SWNTs, whereas they occurred under conditions of 15≤n≤24 for defective SWNTs. Regarding SWNTs with an SW defect, one of the two generated kinks tended to occur on the surface opposite to the defect position, as shown in Figure 1e.

When *n* was further increased, another type of shell buckling mode was observed, which was termed as the “*I*-shaped” mode. Figure 1c,f show the *I*-mode of the (24,0) defect-free SWNT and the (28,0) defective SWNT, respectively. The edge-to-edge length immediately after buckling was estimated to be 12.27 nm (defect-free) and 12.42 nm (defective). Further, the *I*-mode experienced neither columnar bending nor kinks; instead, it was endowed with a fin structure comprising two adjoining flattened areas that were oriented perpendicular to each other. In addition, the *I*-mode was observed when 24≤n≤34 for defect-free SWNTs, whereas 25≤n≤34 was observed for defective SWNTs. Thus, the fin-like necking was expected to occur at the defect position (see Figure 1f).

The armchair and chiral SWNTs also exhibited the above-mentioned series of buckling modes provided the aspect ratio remained unchanged [18]. For instance, previous studies [31,46] revealed that the *I*-mode appeared in (10,10) and (11,11) armchair defect-free SWNTs; the tube lengths were set to be 9.6 and 10.1 nm, respectively, whose aspect ratios were close to those of (24,0) of the proposed zigzag SWNT model with a length of 12.78 nm. However, despite the similarities in buckling modes, there a significant difference that exist between zigzag SWNTs and the other two types of SWNTs in terms of the degree of reduction in the critical buckling stress caused by SW defect insertion, as mentioned in Section 1.

## 4. Force-Strain Curve

Figure 2a shows the relation between the applied compressive force *F* and the resulting strain ε of SWNTs: the defect-free (upper panel) and defective (lower) cases are displayed. In defect-free SWNTs, the axial force and strain were proportional until the axial force reached the maximum value. The larger the value of *n*, the larger the gradient of the force-strain curve. Upon exceeding the maximum point, the force suddenly dropped off and the defect-free SWNTs transformed into one of the three buckling modes described in Section 3. Thereafter, in the buckled states, the applied force remained approximately constant with an increase in the strain, except for the S-mode of defect-free SWNTs observed at 10≤n≤13. In contrast, in the S-mode, the further applied force resulted in a secondary drop immediately after the strain was increased to a certain value (0.037 for the (12,0) defect-free SWNT). Moreover, upon exceeding the secondary threshold, the local surface curvature of the smoothly curved SWNTs in S-mode increased significantly at specific positions, resulting in a local kink similar to those observed in Z-mode. Such a two-step post-buckling deformation in S-mode was not observed in the other two buckling modes of defect-free SWNTs.

In case of SWNTs containing a SW defect, the force-strain curve for the S-mode (i.e., 10≤n≤14) are different from the defect-free case. They exhibit a smooth arc, rather than a sharp peak, at the maximum point, as demonstrated in Figure 2c. Moreover, the disappearance of the sudden drop in the force-strain curve was because the defective SWNTs cannot maintain the initial straight cylindrical shape—even in the elastic region. Prior to reaching the maximum force, they deform slightly into an S shape in a gradual manner, which renders the defining of the critical buckling point associated with the S-mode of defective SWNTs as non-trivial. However, this study defined the S-mode buckling point for defective SWNTS considering the maximum point of the applied force, despite the disappearance of a sharp peak. The buckling of Z-mode and I-mode continued exhibiting discontinuous behavior both in the slope of force-strain curves and deformation, even for defective SWNTs.

Figure 2b is a diagram of the critical buckling force $F_{c}$ and critical buckling strain $\varepsilon_{c}$ of zigzag SWNTs with various index *n*. With an increase in *n* from n=10, the two critical quantities increased until *n* reached 16 or 17 (depending on the presence of a SW defect). A further increase in *n* resulted in the reduction in $\varepsilon_{c}$ and complex (non-monotonic) fluctuations in $F_{c}$, as depicted in the diagram. This feature was applicable overall regardless the presence of an SW defect. Further, the maximum value of $\varepsilon_{c}$ was smaller for defective SWNTs (∼0.042) than defect-free SWNTs (∼0.052). It will be shown later that the maximum values of both $F_{c}$ and $\varepsilon_{c}$ that the SWNT can withstand are associated with the Z-mode buckling state. Thus, SWNTs showing Z-mode buckling with a moderate diameter have the strongest ability to resist buckling deformation.

## 5. Phase Boundary for Buckling Mode Switching

Figure 3a,b show the modulation in $F_{c}$ and $\varepsilon_{c}$ with increasing *n*, respectively. For defect-free SWNTs, the curves of $F_{c}$ and $\varepsilon_{c}$ exhibited two cusps. The left cusp was located near the phase boundary that separated the S- and Z-mode phases. Whereas, the right cusp was located near the boundary between the Z- and I-mode phases. The occurrence of the two cusps can be attributed to the curves of $F_{c}$ and $\varepsilon_{c}$ being composed of the three downward-convex parabolic curves that are depicted by dotted curves in Figure 3. Each parabolic curve corresponded to one of the three different buckling modes. Consequently, among the three buckling modes, the mode with the smallest force and strain required to generate buckling was realized.

For defective SWNTs, the two cusps were rounded and then merged into a single broad peak, whereas the position of the phase boundary appeared insensitive to the insertion of an SW defect. In particular, at the Z- and I-mode phases, the magnitudes of $F_{c}$ and $\varepsilon_{c}$ were reduced by approximately 10–20 and 15–25%, respectively, compared with those for defect-free cases.

## 6. Normalization of Axial Force Applied

Because the compressive force being dealt with is an extensive variable, the larger the tube diameter, the larger the force proportionally. Thus, to eliminate this volumetric effect and obtain a material-specific property, the compressive force is usually divided by the cross-sectional area of the tube, resulting in pressure *p*. However, in the case of SWNTs, there is no unique way to define the wall thickness because it is composed of a single monatomic layer [47]. Therefore, to avoid the ambiguity of the definition of the wall thickness *h*, the applied force *F* was divided by the perimeter length πd of the cylinder’s cross section, which is referred to as the normalized force *f*: (1)$$f=\frac{F}{\pi d}.$$
The subsequent discussion considers *f* as an alternative measure to *F* and *p*, considering that *f* has the same meaning as pressure *p* if it were further divided by the wall thickness *h* of SWNTs.

Figure 4a,c shows the relation between the normalized force *f* and strain ε; the upper (lower) panels correspond to defect-free (a single SW defect induced) SWNTs, respectively. It is evident that within an elastic region, almost all data points are located on a slanted straight line, similar to that in stress-strain curves for macroscopic elastic materials. The slope of the slanted lines estimated from Figure 4a,c are 255–325 N/m and 257–321 N/m, respectively. The similarity in the slopes indicates that Young’s modulus of SWNTs are independent of the presence or absence of an SW defect, which is consistent with the conclusions arrived at by a tight-binding molecular dynamics simulation [48].

Figure 4b,d depict the diagram of critical buckling normalized force $f_{c}$ and the corresponding strain $\varepsilon_{c}$ for defect-free and defective SWNTs, respectively. Almost all data points were observed on a common straight line, except for a few left-bottom data points (colored in red and magenta) that correspond to the S-mode buckling. Comparisons of the two diagrams revealed that the insertion of an SW defect resulted in reduction in the maximum values of $f_{c}$ and $\varepsilon_{c}$ by approximately 19 and 24 %, respectively (represented by the data point located in the upper right corner in the diagram).

## 7. Diameter Dependence of $f_{c}$

### 7.1. Logarithm Plot of $f_{c}\left(d\right)$

As mentioned in Section 1, the buckling performance of SWNTs under compression should be strongly dependent on the aspect ratio of the tube geometry. Consequently, the value of $f_{c}$ deduced from the proposed SWNT model both with and without an SW defect is expected to exhibit strong tube-diameter dependence, as demonstrated below.

Figure 5 shows the logarithm plot of the critical buckling normalized force $f_{c}\left(d\right)$ as a function of tube diameter *d*. At small *d*, the slope of the curve is nearly equal to 2, indicating a nearly square power law of $f_{c}\propto d^{2}$. However, a further increase in *d* resulted in a decreasing tendency of $f_{c}$, while the limit appeared to converge to a constant value. The two contrasting behaviors of $f_{c}\left(d\right)$ at small and large *d* are explained in the following discussion.

### 7.2. Square Power Law of $f_{c}$ at Small d

The square power law of $f_{c}\propto d^{2}$ at small *d* region is the manifestation of columnar buckling described by Euler’s formula. The formula states that a sufficiently slender elastic material under the periodic boundary condition at both ends will buckle at the critical buckling force $F_{c}^{\mathrm{col}}$ expressed by
(2)$$F_{c}^{\mathrm{col}}=\frac{4\pi^{2}}{L^{2}}EI,$$
where *E* is the modulus of elasticity, *I* is the second moment of area of the cross section, and *L* is length of the slender material. The application of the formula to a thin hollow cylinder with diameter *d* and wall thickness h(≪d) yields the associated normalized force $f_{c}^{\mathrm{col}}$ as
(3)$$f_{c}^{\mathrm{col}}=\frac{F_{c}^{\mathrm{col}}}{\pi d}=\frac{\pi^{2}Eh}{4}\cdot \frac{d^{2}}{L^{2}},$$
where the following approximation was used
(4)$$I=\frac{\pi }{64} [{\left(d+\frac{h}{2}\right)}^{4}-{\left(d-\frac{h}{2}\right)}^{4}]≃\frac{\pi }{16}hd^{3}\;at\;\;\frac{h}{d}\ll 1.$$
Equation (Equation 3) is consistent with the square power law of $f_{c}\propto d^{2}$ at small *d*, as observed in Figure 5a.

### 7.3. Decreasing Behavior of $f_{c}$ at Large d

The decreasing trend of $f_{c}$ at large *d* can be plausibly explained from the two different perspectives. First, it is deduced from the continuum elastic theory that an elastic hollow tube does not undergo archwise column buckling when *d* is not extremely small compared to *L*. Rather, it exhibits a circumferential shell buckling characterized by the periodic modulation of the tube diameter along the axial and circumferential directions [32,49]. This type of buckling mode is accompanied by the change in the cross-section shape with the *m*-fold symmetry (*m* is an integer larger than 1). The threshold compressive force $F_{c}^{\mathrm{shl}}$ for the circumferentialshell buckling of mode *m* is expressed as
(5)$$F_{c}^{\mathrm{shl}}=\frac{2\pi Eh^{2}}{\sqrt{3(1-\nu^{2})}}\cdot \frac{m^{2}-1}{m^{2}+1},$$
which leads to
(6)$$f_{c}^{\mathrm{shl}}=\frac{F_{c}^{\mathrm{shl}}}{\pi d}=\frac{2Eh^{2}}{\sqrt{3(1-\nu^{2})}\cdot d}\cdot \frac{m^{2}-1}{m^{2}+1}.$$
The present cases (the Z and I modes) are corresponding to the 2-fold buckling mode (m=2). The axi-symmetric shell buckling is regarded as the extreme case of circumferential buckling mode of m→∞ [1,50].

The second perspective is based on the negative correlation between the bending rigidity of a rolled sheet and its tube diameter. As intuitively understood, rolling a flat paper renders bending it harder; more precisely, rolling a soft flat paper into a cylinder provides the paper, which was originally unable to stand on its own, with sufficient bending rigidity to stand on its own. This phenomenon is also applicable to graphene sheets. Thus, when the diameter *d* of the SWNT formed by rolling a flat graphene sheet is gradually increased, the rigidity *D* against the axial compression decreases gradually. Finally, it converges to the bending rigidity of the flat graphene sheet under in-plane compression. In addition, this negative correlation between *d* and *D* may contribute to the decreasing behavior of $f_{c}$ at large *d*, as observed in Figure 5a.

### 7.4. Approximate Curve for $f_{c}\left(d\right)$

Summarizing the discussion so far, the *d*-dependence of $f_{c}$ observed in Figure 5 can be approximated by the following expression [51]: (7)$${ [\frac{1}{f_{c}\left(d\right)}]}^{\alpha }={ [\frac{1}{f_{c}^{\mathrm{col}}\left(d\right)}]}^{\alpha }+{ [\frac{1}{f_{c}^{\mathrm{shl}}\left(d\right)}]}^{\alpha },\;f_{c}^{\mathrm{col}}\left(d\right)=Ad^{2},\;\;f_{c}^{\mathrm{shl}}\left(d\right)=Bd^{-1},$$
with the proportionality constants A and B defined in Equations (Equation 3) and (Equation 6), respectively. When *d* is sufficiently small, the first term in the rightmost side of Equation (Equation 7) should be dominant and thus $f_{c}\propto d^{2}$ holds. In contrast, if *d* is sufficiently large, the second term becomes dominant and thus $f_{c}$ becomes a decreasing function of *d* as $f_{c}\propto d^{-1}$. These two asymptotic behaviors of $f_{c}$ at large and small *d* imply the existence of a maximum of $f_{c}$ at a moderate value of *d*, as demonstrated in Figure 5a.

The estimation of the constants A and B requires the numeric values of Eh and $Eh^{2}$, as well as ν, for SWNTs. However, the manner in which to properly define the values of these mechanical constants in SWNTs has been a long-standing problem. Many theoretical consequences have proposed that these values are not ideally constants—rather, they can vary depending on the tube diameter. Among the several theoretical suggestions, this study employed the numeric data of *E* and *h* obtained through MD simulations in Ref. [47]. The approximation function of the *h*-*d* and ν-*d* relationships are provided in a previous study, respectively, as follows: (8)$$\begin{array}{ccc} \nu (d) & = & -0.314\ln\left(\frac{d}{2}\right)+0.307, \end{array}$$(9)$$\begin{array}{ccc} h(d) & = & 0.13749-0.18515\exp\left(-\frac{0.5156d}{2a}\right)\;\left[\mathrm{nm}\right], \end{array}$$
where *a* denotes the C-C bond length in SWNTs. However, no approximation function is available for the *E*-*d* relationship (although the numerical simulation data was provided in Ref. [47]). Thus, this study attempted to develop an approximation function. The Eh-*d* relationship was fit instead of directly fitting the *E*-*d* relationship itself for the following two reasons: (i) the Eh product (also referred to as “2D elastic modulus”) can be uniquely determined from the normalized force-strain (*f*-ε) curve; and (ii) with increase in *d*, the Eh product is expected to converge quickly to a certain value (2D elastic modulus of graphene). Consequently, these features render the approximation process easier, and a sufficiently simple function form can be presumed.

Figure 5b shows the values of the product Eh as a function of *d* deduced from Ref. [47]; the solid curve is an exponential fitting curve defined by
(10)$$Eh=c_{1}\exp\left(-\frac{d}{c_{2}}\right)+c_{3},$$
where the appropriate values of the fitting parameters, $c_{i}$(i=1,2,3) were calculated by employing a nonlinear least square method. All the data points were found to be well fitted by Equation (Equation 10) with $c_{1}=13.3$ N/m, $c_{2}=0.907$ nm, and $c_{3}=331$ N/m. The *E*-*d* relationship can be obtained by dividing Equation (Equation 10) by Equation (Equation 9). Based on this, the values of A and B were evaluated for different *d*s, followed by substituting them into Equation (Equation 7) to obtain the approximate curve of $f_{c}\left(d\right)$. The solid lines shown in Figure 5a is the obtained approximate curve with setting of α=4.0. The square power law at small *d* was accurately reproduced by the curve. At large *d*, the curve was fairly consistent with the data points, even though Equation (Equation 7) was originally introduced for prediction of the buckling behavior in the continuum bodies.

It is noteworthy that the influence of the model dimensions on the buckling behavior is expected to differ between thick and thin SWNTs. Thus, change in the SWNT model size, such as length, can lead to different buckling behaviors. Indeed, how the SWNT model size affects the buckling behavior can be estimated from Equations (Equation 3) and (Equation 6). In the case of thin SWNTs (i.e., small aspect ratio, d/L), Euler’s buckling law of Equation (Equation 3) holds approximately. In this case, the buckling behavior is governed by the single parameter of the aspect ratio, d/L, rather than being influenced by *d* and *L* independently. In the case of thick SWNTs, on the other hand, the effect of the model size is given only by *d*. The above consideration leads to the possibility that the boundaries between the S, Z and I buckling modes may change with varying the model size.

### 7.5. Defect-Induced Reduction in $f_{c}\left(d\right)$

Figure 5c shows the reduction in the critical buckling normalized force $f_{c}$, defined by the ratio of
(11)$$\frac{f_{c}^{\mathrm{SW}}-f_{c}^{\mathrm{Id}}}{f_{c}^{\mathrm{Id}}},$$
where $f_{c}^{\mathrm{SW}}$ and $f_{c}^{\mathrm{Id}}$ are the values of $f_{c}$ in a defective and defect-free (i.e., ideal) SWNT for a particular chiral index *n*, respectively. This ratio accounts for the $f_{c}$-reduction caused by the insertion of an SW defect into the pristine SWNT model. Figure 5c shows that an SW-defect insertion results in a reduction in $f_{c}$ by approximately 10–25% depending on *n*. In particular, the magnitude of reduction in $f_{c}$ was enhanced at n=16 and n=22; these values of *n* coincided with the tube diameters *d* at which $f_{c}$ for the defect-free system showed upward sharp cusps as observed in Figure 5a. This finding implies that the SWNT, which shows the strongest buckling resistance in the absence of SW defects, exhibits the most significant reduction in the buckling resistance when a SW defect is inserted.

## 8. Buckling of SWNTs with Two SW Defects

Intuitively, it is expected that the larger the number of defects contained in a system, the smaller the value of the critical buckling force of the system. However, this expectation is not always applicable to SWNTs having SW defects, as demonstrated below.

Figure 6b shows the force-strain curve for (10,0) zigzag SWNTs having two SW defects. The defect position varied as depicted in Figure 6a, wherein the SW defect indicated by the bottom red square was fixed but the one indicated by the upper red square was shifted step-by-step along the direction parallel to the circumference of the tube. Regarding the defect configurations labeled by H0 and H1, the maximum point of the force-strain curve exceeded the maximum for the single-SW case, implying that the SWNT having two defects can withstand a compressive force larger than the SWNT having only one defect. However, such a counter-intuitive phenomenon does not occur for the defect configurations shown in Figure 7a, where the two SW defects are always located on opposite sides of the cylindrical axis of the (10,0) zigzag SWNT. In the latter case, the maximum value of the force-strain curve for two-defect systems were always below that for a single-defect system as shown Figure 7b, regardless of changes in the distance between the two SW defects.

The resistance to the axial compression increased with the increase in the number of SW defects because the buckling mode that occurred in (10,0) zigzag SWNT was always in the S-mode. In the S-mode buckling, the two positions at which the Gaussian curvature of the deformed cylindrical surface acquired the maximum positive value were always located on opposite sides of the cylindrical axis. If an SW defect was present at each of these two positions, the original straight shape of the SWNT became slightly unstable under compressive force because the carbon atoms near the SW defect preferred to be displaced in the out-of-plane direction. This is exactly the case realized in the defect configuration labeled by V15 in Figure 7a. However, if one of the two SW defects located at the position where the surface curvature exhibited a negative maximum value, as shown in the configuration labeled by H0 in Figure 6a, the out-of-plane displacement of the SW defect tended to suppress the archwise deflection of the tube that locally promoted the negative Gaussian curvature of the surface.

Thus, increasing the number of SW defects may increase the mechanical resistance of the SWNT to the compressive load, depending on the relative configuration between the positions where the surface Gaussian curvature is maximum in the buckled states and the position where the SW defect exists.

## 9. Summary

This study investigated the axial buckling of zigzag SWNTs having SW defects while changing the value of the tube diameter *d*. The effect of changing the tube diameter *d* was manifested primarily in the transition between the three buckling modes (S, Z, and I) and the contrasting behavior of the critical buckling load at large and small *d* regions. The square power law at small *d* was due to the Euler-type columnar buckling, whereas the decreasing trend at large *d* was explained by considering the local shell buckling mode. In addition, it was found that the inclusion of multiple SW defects in zigzag SWNTs resulted in the counterintuitive phenomenon of the buckling resistance being strengthened with an increase in the number of defects increases.

## Figures and Tables

**Figure 1 nanomaterials-12-02617-f001:**
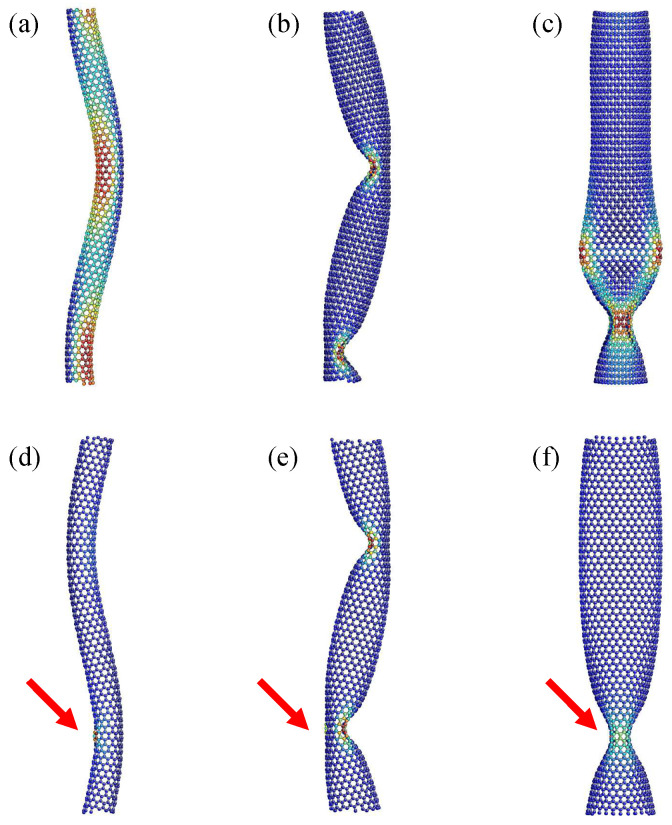
Three buckling modes of zigzag SWNTs with no defect (upper panels) and with a single SW defect (lower): (**a**,**d**) S-mode, (**b**,**e**) Z-mode, (**c**,**f**) I-mode. The red arrows indicate the locations of the SW defect. The chiral indices of SWNTs are: (**a**,**d**) (12,0), (**b**,**e**) (18,0), (**c**) (24,0), and (**f**) (28,0).

**Figure 2 nanomaterials-12-02617-f002:**
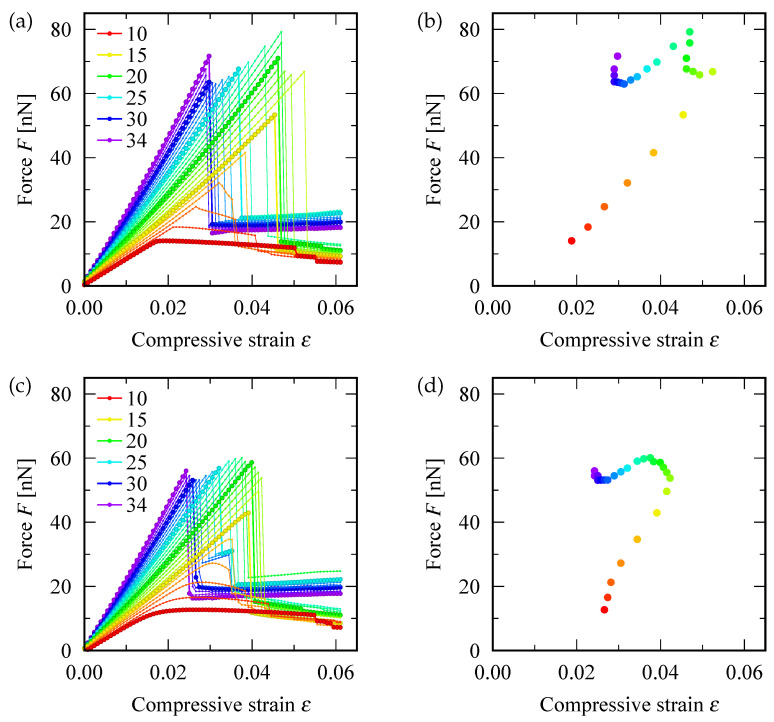
(**a**,**c**) The force-strain curve (*F* vs.ε) of zigzag SWNTs under axial compression, and (**b**,**d**) diagram of the critical buckling force $F_{c}$ and critical buckling strain $\varepsilon_{c}$. The top (bottom) panels correspond to pristine (defective) SWNTs. The difference in the color of data points represents the value of the chiral index *n* shown in the legend of the panels (**a**,**c**); i.e., the red (n=10) and purple (n=34) lines are corresponding to the thinnest and thickest SWNTs, respectively.

**Figure 3 nanomaterials-12-02617-f003:**
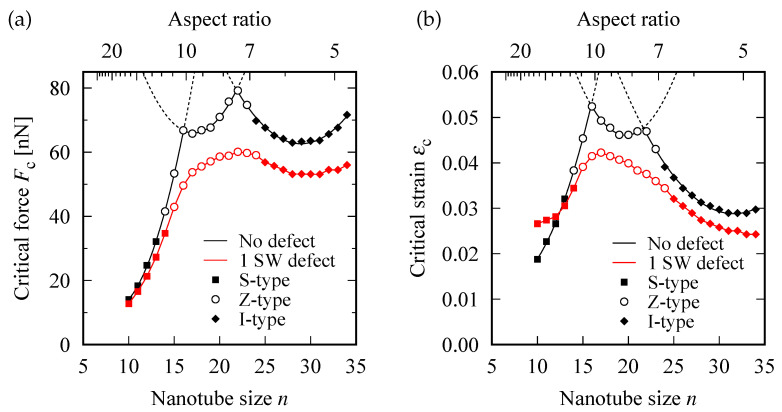
Tube diameter-driven modulations in: (**a**) the critical buckling force $F_{c}$, and (**b**) critical buckling strain $\varepsilon_{c}$. The presence/absence of a SW defect and the buckling mode occurred are distinguished by differences in colors and symbols, respectively. The curve profile with two cusps in the data of the defect-free models is fitted with three parabolas each (solid and dashed lines), which are corresponding to the S, Z, and I buckling modes.

**Figure 4 nanomaterials-12-02617-f004:**
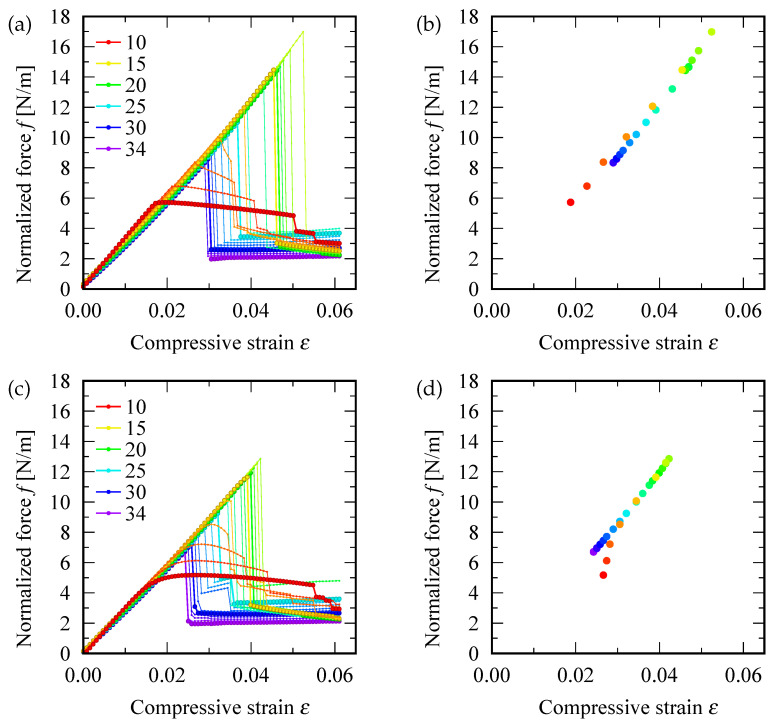
(**a**,**c**) Normalized force f≡F/(πd) (i.e., applied compressive force *F* divided by circumferential length πd) as functions of the compressive strain ε. (**b**,**d**) Diagram of the critical buckling normalized force $f_{c}\equiv F_{c}/\left(\pi d\right)$ and critical buckling strain $\varepsilon_{c}$. The top (bottom) panels correspond to pristine (defective) SWNTs. The colors of data points and lines represent the chiral index *n*.

**Figure 5 nanomaterials-12-02617-f005:**
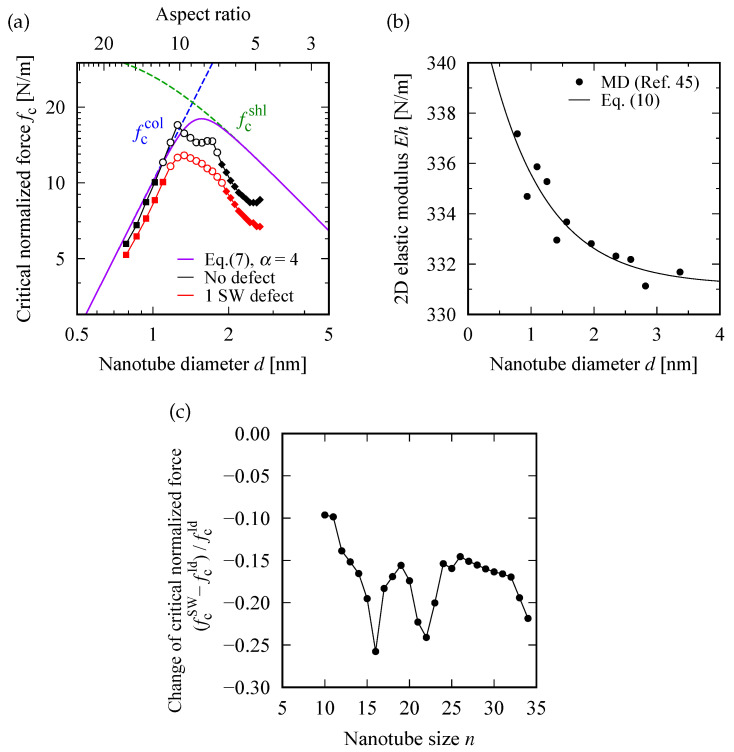
(**a**) Logarithm plot of $f_{c}\left(d\right)$ as a function of tube diameter *d*. Two dashed curves, labeled by $f_{c}^{\mathrm{col}}$ and $f_{c}^{\mathrm{shl}}$, are deduced from Equations (Equation 3) and (Equation 6), respectively. The solid curve corresponds to the approximated expression expressed as Equation (Equation 7) with the parameter setting of α=4.0. (**b**) Numeric data of Eh obtained by MD simulations published in Ref. [47]. An exponential fitting curve to the data point, calculated by a nonlinear least square method, is depicted by a solid curve; the explicit functional form is provided in the text. (**c**) Reduction in the critical buckling normalized force caused by insertion of a SW defect; $f_{c}^{\mathrm{SW}}$ and $f_{c}^{\mathrm{Id}}$ are the value of $f_{c}$ for defective and defect-free SWNTs, respectively.

**Figure 6 nanomaterials-12-02617-f006:**
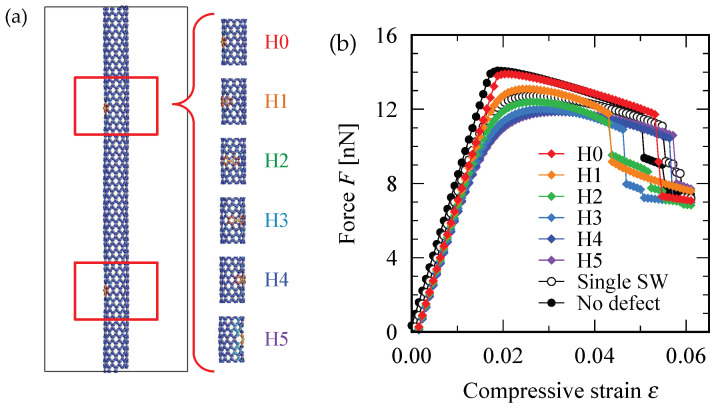
(**a**) Configuration of the two SW defects. The defect position is shifted in the direction horizontal to the circumference of the tube. (**b**) Force-strain curve for the different defect configurations. The critical point for the defect configurations labeled by H0 and H1 exceed the one-SW-defect curve, indicating the paired-defect-induced enhancement in the resistance to axial compressive force.

**Figure 7 nanomaterials-12-02617-f007:**
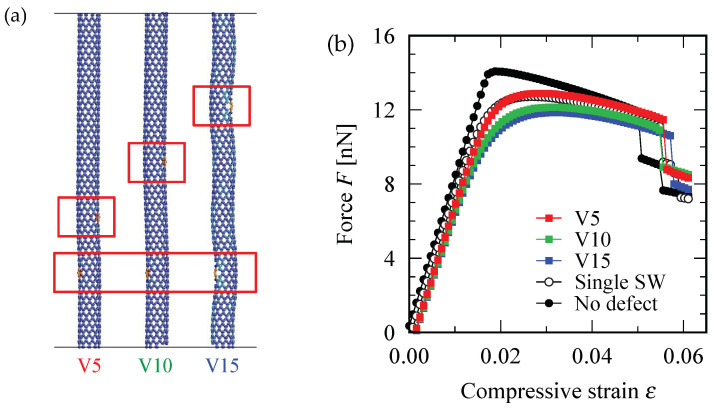
(**a**) Configuration of the two SW defects. The defect position is shifted in the direction vertical to the circumference of the tube. (**b**) Force-strain curve for the different defect configurations. The critical points of almost all defective curves are below that of the one-SW-defect curve.

## Data Availability

The data presented in this study are available on request from the corresponding author.

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
