# Peer review of "Diameter-Change-Induced Transition in Buckling Modes of Defective Zigzag Carbon Nanotubes"

_nanomaterials, 2022, doi:10.3390/nano12152617_

Round 1
Reviewer 1 Report
Umeno and coworkers have studied the buckling of zigzag SWCNT under axial load and the effect of SW defects. The methods and findings are clearly described.
I have only one small question. How would the S,Z,I buckling modes depend on the length of the simulation cell? e.g. if it was simply doubled. From the current discussion I would presume that Z and I in Fig.3 should not be affected by doubling the length, but the curve for the S mode should shift to larger diameters. Would there by a critical maximum length for the direct Z buckling?
Otherwise I recommend publishing after minor grammer polishing.
Reviewer 2 Report
Recommendation: This paper is not recommended in current version.
Comments:
In this manuscript, the authors performed MD simulations for the buckling performance of the zigzag-type SWNTs that depend on various tube diameters under the free defects or Stone-Wales (SW) defects situation. They reported that the relation between the force-strain curve and aspect ratio of the zigzag SWNTs with SW defects, as well as that of the free defects tubes for comparison. Although this work appears to have been carefully performed, this paper offered no some new or interesting insights into the structure and mechanical properties of SWNTs under the common axial compression, and the content is not sufficient to support their viewpoints. Besides, the sentences and captions were incomplete organized in this paper, which makes it difficult to understand for reader, and this should be carefully checked and revised before resubmission. On the whole, the paper is not acceptable in the current form. Some more specific comments are listed below:
Comment 1: The whole paper reading likely there is no substance content, which the parts for the Figure 3 and Figure 4 content seemingly repeat from Figure 2 in other way, the three are essentially no different. The authors should deeply reveal the reasons instead of more looking at the picture, such as the reasons for the relation between the force-strain properties and the tube index n. Can the effect observed be understood within the picture of energy landscape?
Comment 2: In this work, the authors intensively claim that they obtained an approximate expression for the critical buckling force as a function of the tube diameter was formulated. However, they mentioned the expressions are obtained from other references. Thus, these are not the original work and the interesting insights, such content are suggest not the focus of this article.
Comment 3: What is the standard for distinguishing the three buckling modes depending on the tube diameter? What is the small d, the large d? Why the difference is so obvious in the S-mode of zigzag SWNTs with no defect and with a single SW defect? In Section 8, please explain the reason for the relation between the defect position and the mechanical properties of SWNTs under compression.
Comment 4: The simulation method is written too simple to unclear, such as why choose the temperature at 50 K? What is the influence under the room temperature case?
Comment 5: Some obvious writing errors can be found in this manuscript, such as the Figure 1 caption “(a,b) (12,0), (b,e) (18,0)”.
Comment 6: In Figures 2 and 4, the authors didn’t explain the meaning of each curve lines in detail under the Figures’ captions, and the dashed lines of the fitted with three parabolas in Figure 3 are so confused. The authors should go through check the manuscript carefully.
Reviewer 3 Report
The manuscript of Umeno et al. reports the data of MD simulations using a reactive force field method on the mechanical properties of single-walled zigzag nanotubes under axial load. The authors study dependencies of critical forces, critical strains and scenarios of nanotubes’ deformation on the chiral number of nanotubes and the presence of a possible Stone-Wales defect. The found characteristics are fitted to analytical expressions derived earlier for continuum bodies. The subject of the study belongs to the topics of Nanomaterials in section Theory and Simulation of Nanostructures. I could recommend this manuscript for publication after the authors consider next critical comments and make appropriate additions to the manuscript.
1) Lines 68-69, the authors write “…with an initial length of 12.78 nm; along the tube axis, 60 hexagonal carbon rings were lined up from end to end.” Why 12.78 nm has been chosen? Was it optimized using REBO force field value? Was it found the same for all nanotubes irrespective their chiral number? I doubt that it will be one and the same value for different nanotubes. If it is really different, then it seems that some nanotubes have been be pre-stressed at initialization of MD simulations and the results of this study can be different.
As well I do not understand well the term “60 hexagonal carbon rings” – did the authors mean 60 unit cells composing the supercell of a SWNT?
2) Part 2.2 describes MD procedure only briefly. How many MD stages were used to reproduce a deformational curve? How the atom velocities were set up at the first MD stage and at the following MD steps? How many MD steps were used per a MD stage? What was MD time step? Which kind of software was employed? At the moment, this study cannot be reproduced by other researchers.
3) Lines 91-92, I find MD run for 10 fs as too short and the temperature of 50 K as too low. It may happen that such multi-atom models did not fall into the global minimum state.
4) How many times the deformation curves were reproduced for every nanotube? How large is deviation of the forces’ values? It can be mentioned in value in the text or given as error bars on a figure.
5) Lines 308-309, the authors wrote “This finding implies that the SWNT with the strongest resistance to compressive force is the SWNT with the weakest resistance to a SW-defect insertion.” It is not clear, where does the “resistance to a SW-defect insertion” come from. I guess, it is was not studied here.
6) What is fundamentally new is presented in the study, when comparing to the paper [18] from 2018? The same authors have used the same method, the models of both zigzag and armchair SWNTs and had described the same “S,Z,I” scenarios of axial loading already there.
Round 2
Reviewer 2 Report
Accpet.
Reviewer 3 Report
The authors have considered all my critical comments and have revised the manuscript accordingly. I recommend this manuscript for acceptance.